# Transparent and Flexible SiOC Films on Colorless Polyimide Substrate for Flexible Cover Window

**DOI:** 10.3390/mi12030233

**Published:** 2021-02-25

**Authors:** Jin-Hyeok Park, Chan-Hwi Kim, Ju-Hyeon Lee, Han-Ki Kim

**Affiliations:** School of Advanced Materials Science and Engineering, Sungkyunkwan University, Suwon-si, Gyeonggi-do 16419, Korea; parangse94@skku.edu (J.-H.P.); wwe1126@g.skku.edu (C.-H.K.); wngus2se@g.skku.edu (J.-H.L.)

**Keywords:** SiOC, hard coating films, colorless polyimide, flexible and foldable displays, roll-to-toll sputtering

## Abstract

We fabricated transparent and flexible silicon oxycarbide (SiOC) hard coating (HC) films on a colorless polyimide substrate to use as cover window films for flexible and foldable displays using a reactive roll-to-roll (R2R) sputtering system at room temperature. At a SiOC thickness of 100 nm, the R2R-sputtered SiOC film showed a high optical transmittance of 87.43% at a visible range of 400 to 800 nm. The R2R-sputtered SiOC films also demonstrated outstanding flexibility, which is a key requirement of foldable and flexible displays. There were no cracks or surface defects on the SiOC films, even after bending (static folding), folding (dynamic folding), twisting, and rolling tests. Furthermore, the R2R-sputtered SiOC film showed good scratch resistance in a pencil hardness test (550 g) and steel wool test under a load of 250 g. To test the impact protection ability, we compared the performance of thin-film heaters (TFHs) and oxide-semiconductor-based thin-film transistors (TFTs) with and without SiOC cover films. The similar performance of the TFHs and TFTs with the SiOC cover window films demonstrate that the R2R-sputtered SiOC films offer promising cover window films for the next generation of flexible or foldable displays.

## 1. Introduction

The rapid advancement in flat panel display (FPD) technologies has made it possible to realize flexible displays for next-generation consumer electronics, such as mobile phones, tablet PCs, and TVs. Due to the requirements of consumers and the development of FPD technologies, the form factor of display products has changed from a rigid flat panel-type to flexible and deformable panel types. Flexible displays can be divided into several categories, such as curved, rollable, foldable, and stretchable displays, depending on the curvature and deformation mode of substrate [1,2,3]. Recently, foldable displays have received great attention in state-of-art mobile display due to their unique design, light weight, and large area screen [4,5,6,7]. For the commercialization of foldable displays, it is necessary to develop key components, such as foldable organic light-emitting diodes (OLED), foldable thin-film transistors (TFTs), high-performance plastic substrates to replace rigid glass substrates, and transparent and flexible cover window for flexible displays [8]. In addition, for the mass production of flexible cover windows, transparent hard coating (HC) films should be coated on flexible window substrates using a roll-to-roll (R2R) coating process [9,10,11]. Although extensive research on flexible OLEDs, flexible TFTs, and plastic substrates has been reported [12,13,14,15,16], investigation of transparent and flexible cover window films that are appropriate for foldable displays is still lacking. In particular, the foldable display operates at a small bending radius or severe curvature, so the flexible and transparent coating films on cover window films should be mechanically flexible. The main role of cover window films in foldable displays is the protection of the foldable display panels from external impacts and scratches. In the case of glass-based FPDs, the cover glass could easily protect the display panels from external impacts and/or scratches due to the hardness of the cover glass. However, in foldable displays the development of transparent and flexible HC films coated on window substrates is required for the protection of flexible display panels. Hwang et al. reported a ladder-like polysilsesquioxane (LPSQ) HC film prepared by an ultraviolet (UV) curing process [17]. While the scratch resistance of the LPSQ film was excellent, the mechanical properties of the flexible device were poor. The transparent and flexible HC films on cover window films need to meet certain crucial requirements, such as high optical transmittance, superior flexibility, high scratch resistance, and high impact protection ability for foldable displays [18]. As mentioned above, it is very important to develop HC materials for a cover window of foldable displays. Among several HC materials, silicon oxycarbide (SiOC) has been considered a promising HC material due to its low dielectric constant, good mechanical flexibility, and superior hardness [19,20]. However, to the best of our knowledge there is no report in the literature of reactive R2R-sputtered SiOC films on colorless polyimide (CPI) substrate for use as transparent and flexible HCs on cover windows.

In this study, we developed novel SiOC HC films on a CPI substrate for the cover window of foldable electronics via a reactive R2R sputtering process. The R2R sputtering process is suitable for the fabrication of flexible HC materials due to its advantages of large scalability, high deposition rate, and controllability of composition. Herein, the reactive R2R sputtering process enables the mass production of large-scale HC materials and the formation of uniform SiOC films on CPI substrates by reactively sputtering with an injection of oxygen gas. We investigated the optical, surface, and mechanical properties of the SiOC HC layers deposited on flexible CPI substrates (SiOC/CPI) using a lab-scale R2R sputtering system. In particular, the mechanical properties of SiOC/CPI were investigated to show its feasibility as a transparent and flexible HC for a cover window in foldable displays. Furthermore, the impact protection ability of the SiOC HC film was investigated by comparing the performance of thin-film heaters (TFHs) and oxide-semiconductor-based thin-film transistors (TFTs) with and without an SiOC cover film. The SiOC/CPI films showed a superior transparency of above 85% and good mechanical flexibility under bending, rolling, folding, and twisting tests at a small curvature of 3 mm. Furthermore, the SiOC/CPI films could endure scratching tests and mechanical pushing tests without the deterioration of the device’s performance.

## 2. Materials and Methods

Figure 1 shows the schematic reactive sputtering process to deposit SiOC HC films at room temperature (RT). The flexible and transparent SiOC films were coated on 50 μm-thick CPI substrates (Kolon Industries, Seoul, Korea) via a reactive sputtering process using a lab-scale R2R sputtering system. The commercial CPI roll of 250 mm width was loaded in an unwind and rewind roller for a continuous R2R sputtering process. The SiOC films were sputtered using a 99.9%-purity silicon carbide (SiC) target (Thifine, Incheon, Korea) with a constant mid-frequency (MF) power (Trumpf, Ditzingen, Germany) of 5000 W. Oxygen was incorporated into the SiOC HC films by the injection of 50 sccm of O_2_ gas with 500 sccm of Ar gas into the R2R sputtering system. As shown in the enlarged dash lines in Figure 1, when the CPI substrate passes the plasma region above the SiC target the SiOC films were coated on the CPI substrate, which is in physical contact with the cooling drum. The thickness of the SiOC films was adjusted by the rolling speed of the CPI substate that was passing over the SiC target. The thickness of the deposited SiOC films on CPI substrates was measured using a surface profiler (Dektak 6M, Veeco, Plainview, NY, USA). The optical transparency of the SiOC films was measured by UV/Visible spectrometry (UV-540, Unicam, Hachioji, Japan). The chemical composition of the SiOC HC films was then studied by X-ray photoelectron spectroscopy (XPS) (ESCALAB 250, Thermo Scientific, Waltham, MA, USA) with an Al–Kα source. To investigate the flexibility of the R2R-sputtered SiOC/CPI, we employed lab-made mechanical flexibility testers, such as a bending (static folding) tester, a folding (dynamic folding) tester, a twisting tester, and a rolling tester. Following that, a pencil harness tester (KP-M5000M, Kipae E&T, Suwon-City, Korea) and steel wool scratching tester (KP-M4250, Kipae E&T, Suwon-City, Korea) were employed to test the surface scratch resistance properties of the R2R-sputtered SiOC films. After the flexibility test and surface hardness test, the surface images of SiOC films were monitored using field emission-scanning electron microscopy (FE-SEM:JSM-7600F, JEOL, Tokyo, Japan) and optical microscopy (OM:DM2700 M, Leica, Wetzlar, Germany). Finally, to investigate the protection ability of SiOC/CPI films, we fabricated thin-film heater (TFHs) with sizes of 25 × 25 mm^2^ and oxide-semiconductor-based thin-film transistors (TFTs). The Ag network were used for the fabrication of TFHs. Ag paste was coated on the side of an Ag network film to form a contact electrode for applying direct current (DC) voltage. Then, the contact electrodes were covered with conductive Cu tape. The temperature and IR thermal image of the TFHs were monitored using a thermocouple and IR camera under a DC voltage of 3 V. SiOC/CPI films were applied to test impact resistance using an impact pushing tester (ET-126-4, Labworks Inc., Costa Mesa, CA, USA). The 10 cycles of impact pushing of 6 N was applied to the TFHs and TFTs which were covered with or without SiOC/CPI films. TFTs with a conventional bottom-gate configuration were fabricated on a SiOx/p^++^ Si substrate. Here, 200 nm-thick SiO_x_ and p^++^ Si are served as a gate insulator and a global gate electrode. On the SiOx/p^++^ Si, a 30 nm-thick indium zinc tin oxide (IZTO) channel layer was deposited by the radio frequency (RF) sputtering of IZTO (99.99% purity, In:Zn:Sn = 1:1:1 at.%) target, and a 100 nm-thick indium tin oxide (ITO) source/drain (S/D) electrode layer was deposited using the DC sputtering of ITO (99.99%, In:Sn = 90:10 wt.%) target. Both the channel and S/D patterns were defined via invar metal shadow masks. Finally, the I-V characteristics of the TFTs were measured using a semiconductor parameter analyzer (4200-SCS, Keithley, Cleveland, OH, USA). After the mechanical pushing test, the performance of the TFHs and TFTs was compared with or without the SiOC/CPI cover windows.

## 3. Results and Discussion

To investigate the binding state and composition of the R2R-sputtered SiOC HC film on the CPI substrate, an XPS depth profile and survey analysis was conducted. Figure 2a represents the atomic composition of the SiOC/CPI as a function of etch time from surface to substrate. The blue shadow region in the XPS depth profile indicates the SiOC layer on the CPI substrate. In the SiOC layer, the atomic percentages of each atom, such as Si, C, and O, were maintained uniformly with the increasing etching time. The uniform atomic composition of SiOC HC implies that, during the reactive sputtering of the SiC target, the oxygen atoms are incorporated well into the SiOC films. However, because of the organic contamination on the surface of the SiOC film, the atomic percentage of C atom is relatively higher at the surface region of SiOC than at the SiOC film region. Due to the residual C-based gas products, such as CO_2_ and CO, that are formed during the reactive sputtering process using a SiC target, it was found that the number of C atoms in SiOC films is much lower than that of Si atoms. Figure 2b–d show the XPS spectra at the core level of Si 2*p*, C 1*s*, and O 1*s* obtained from the SiOC film, respectively. The XPS spectra were calibrated with the center of the C 1*s* peak at 284.5 eV. Figure 2b shows that the XPS spectra of Si 2*p* are deconvoluted into three Gaussian peaks, which are located at 100.5, 101.6, and 102.6 eV [21] and which indicate the SiO–C_3_, SiO_2_–C_2_, and SiO_3_–C bonding states, respectively.

Most Si atoms exist in combination with oxygen and carbon atoms. In particular, the fraction of Si atoms bonds with only O atoms (Si–O_2_) and C atoms (Si–C), which are located at (98.7 and 103.6) eV, respectively, is negligible. The results of the XPS Si 2*p* spectra imply that when the SiC target is sputtered with the Ar/O_2_ gas ambient, the Si atoms are deposited on CPI substrate in the form of SiO_x_C_y_. In addition, the XPS C 1*s* and O 1*s* spectra are also deconvoluted into subpeaks according to the chemical bonding states in Figure 2c,d [22,23,24]. The XPS C 1*s* spectra for Si–C, C–C, and C–O are deconvoluted into subpeaks that are located at 283.3, 284.5 and 286.2 eV, respectively. Because the XPS survey analysis was performed at the surface of the SiOC HC layers, the fraction of C–C bond is dominant in the XPS C1*s* spectra. Meanwhile, the O atoms mainly bond with Si atoms rather than C atoms, as shown in Figure 2d.

Figure 3 shows the optical transmittance of the SiOC/CPI film and bare CPI substrate. The 100 and 300 nm-thick SiOC films coated on the CPI substrate showed high optical transmittances of 87.43 and 87.17%, respectively, in the visible range of 400–800 nm. Due to the excellent transparency of the SiOC films, the transmittances of 100 and 300 nm-thick SiOC HC-deposited CPI samples were almost the same as those of the bare CPI substrate. In the wavelength range 600 to 1200 nm, the transmittance of the SiOC-deposited CPI is slightly higher than that of the bare CPI substrate, as shown in the inset of Figure 3 [25,26,27].

Additionally, the photographs in Figure 3 demonstrate the high transparency of the SiOC/CPI sample, which is sufficiently transparent to see the university logo behind the samples. Due to the high transparency and flexiblity of the SiOC film, the SiOC-coated CPI could be applied as flexible cover window in flexible and foldable displays. In addition to the superior transparency of SiOC/CPI films, SiOC has been reported to have an index of refraction about 1.4 to 1.8, though it changes depending on the fabrication method and composition [28,29]. Memon et al. reported that the index of refraction of SiOC is affected by the ratio between the Si-C bond area and the Si-O bond area. Additionally, Yoon et al. raised the transmittance of cover window films by adjusting the index of refraction [30]. According to the previous reports, the optical properties of our SiOC/CPI cover window films can be also improved by adjusting the oxygen flow rate when the SiOC films were deposited.

For use as a flexible cover window in foldable display, SiOC/CPI samples need to possess good mechanical flexibility and endure external force. Therefore, we investigated the mechanical flexibility of the SiOC/CPI samples via various mechanical flexibility tests, such as bending (static folding), folding (dynamic folding), twisting, and rolling tests, using lab-made test systems. Figure 4a–d show the bending test steps during the repeated bending, folding, twisting, and rolling tests, respectively, using SiOC/CPI samples. The outer and inner bending tests were repeated at a bending radius from infinity to 2 mm for 100,000 cycles. The bending radius was formed by pressing both sides of the clipped sample, as shown on the lower illustration of Figure 4a. After 100,000 cycles of repeated outer/inner bending tests, we confirmed the bendability of the SiOC/CPI samples through testing for the existence of surface cracks. The field emission scanning electron microscopy (FE-SEM) images of Figure 4a show that there are no surface cracks or defects at the surface of SiOC/CPI samples with 100 and 300 nm-thick SiOC HC layers, even after 100,000 cycles. Identical surface FE-SEM images indicate the outstanding mechanical flexibility of the SiOC films. Figure 4b also shows the repeated folding test steps of the SiOC/CPI samples at a folding radius of 2 mm for 100,000 cycles.

The lower illustration of Figure 4b shows the formation of the folding radius. The folding radius was formed while the clipped sample was folded until 180° on the clamp that had round shape and the desired radius. The following equation indicates the maximum peak strain applied to the SiOC/CPI samples depending on the thickness of films and substrates when the bending/folding radius is fixed [31,32].
Strain (ε) = (*d*_SiOC_ + *d*_CPI_)/2*R*,(1)
where *d*_SiOC_ and *d*_CPI_ imply the thickness of SiOC HC layers of 100 or 300 nm and CPI substrate (50 μm), respectively. Then, *R* in Equation (1) means the radius used in the bending or folding flexibility tests. After the outer/inner folding tests for 100,000 cycles at the folding radius of 2 mm, the 100 nm-thick SiOC/CPI films showed no surface cracks. owever, after the repeating folding tests in the right images of Figure 4b, noticeable surface cracks were evident in the FE-SEM images of the 300 nm-thick SiOC film. This difference between the surface cracks of the 100 and 300 nm-thick SiOC/CPI films can be explained using Equation (1), which indicates that the samples with thicker SiOC films were subjected to larger strains at their surface region. Therefore, after the folding test, the thicker SiOC film showed severe cracks. Figure 4c,d show the procedure steps and surface images after the repeated twisting and rolling tests, respectively. The twisting test was performed by twisting the samples with a maximum twist angle of 20°. Then, the SiOC/CPI samples were rolled by a 3 mm-thick cylindrical bar. The twisting and rolling tests were repeated for 100,000 cycles. Figure 4c,d show that after the twisting and rolling tests, even after the repetition of 100,000 cycles, there are no surface cracks in the FE-SEM surface images. Comparison of all of the results of the previous mechanical flexibility tests confirmed that due to their outstanding flexibility, the SiOC/CPI films can be acceptable as a flexible cover window for flexible or foldable display products.

One of the crucial properties for the flexible cover window is a high scratch resistance. We analyzed the scratch resistance via a pencil hardness tester and steel wool scratching tester. During the pencil hardness test, pencil leads with a standard hardness grade of 3 to 6 H were moved across the surface of SiOC films under the pressure of 550 and 750 g loads. Then, steel wool of #0000 grade was moved on the SiOC samples under weights of 250 g for 500 cycles. Figure 5a shows the surface optical microscopy (OM) images of SiOC films after the scratching of pencil under a special weight, while the inset shows the method of the pencil hardness test. Figure 5a shows that under the weight of 550 g, the SiOC HC layers with thickness of both 100 and 300 nm resist the pencil grade of 6H. However, under the weight of 750 g, the 300 nm-thick SiOC films are scratched at a pencil grade from 3H, while in the case of the 100 nm-thick SiOC film surface scratches begin from a higher pencil grade of 4H. The anti-scratching properties of SiOC films in steel wool test were evaluated using transmittance, contact angle, and OM images. Figure 5b shows the change in transmittance after the steel wool scratching test. After the steel wool test, the transmittance of the SiOC films with thicknesses of 100 and 300 nm was barely changed. However, Figure 5c shows that after the steel wool test, the contact angle of H_2_O droplet with SiOC films increased. In the case of the 100 nm-thick SiOC HC layers, the H_2_O contact angle increased from 44.2 to 66.77°; in the case of SiOC films with a thickness of 300 nm, changed from 40.76 to 69.29°. Herein, the contact angle increased because of the increase in surface roughness induced by the steel wool test [33,34]. This can be explained by the Cassie–Baxter theory, according to which, air molecules that are trapped in the furrows in a rough surface generate a composite (solid/air interface) hydrophobic surface, resulting in a large contact angle in comparison to a flat surface [35,36,37].
cos *θ*_CB_ = *f* − 1 + *f*·cos *θ.*(2)

In Equation (2), *f* represents the fraction of the solid/liquid interface, and *θ*_CB_ means the contact angle by the Cassie–Baxter theory. The degree of increase in contact angle after the steel wool test was higher for the 300 nm thick SiOC HC/CPI films than for the 100 nm thick SiOC HC/CPI films. Moreover, surface damage after the steel wool test was observed on only the 300 nm thick SiOC hard coating films shown in the OM images (insets of Figure 5c). These results indicate that the 100 nm thick SiOC HC has greater scratching resistance. The overall results of the pencil hardness test and steel wool scratching test indicate that the resistance to scratching of the 100 nm thickness SiOC films is better than that of the 300 nm films. In previous reports, thicker films generally showed a higher scratch resistance than thinner films [38,39]. However, in our study, the 100 nm-thick SiOC films showed a higher scratch resistance than the 300 nm-thick SiOC films due to their better adhesion, and the thickness under several micrometers rarely suffered scratching [40].

To demonstrate the impact protection ability of the SiOC/CPI samples as a flexible cover window for flexible and foldable displays, we conducted the impact pushing test. For the impact pushing test, we intentionally fabricated Ag network-based TFHs. The Ag network-based TFHs were prepared as shown in Figure 6a. Ag paste was coated on the side of an Ag network film with dimensions of 25 × 25 mm^2^ to form a contact terminal for applying DC voltage. Finally, the contact region was covered with conductive Cu tape. A DC voltage of 3 V was applied to the TFHs through the Cu contact, and the temperatures and infrared (IR) thermal images of the TFHs were monitored using a thermocouple and IR camera. Figure 6b shows the steps of the impact pushing test. The load of the impact pushing test was fixed at 6 N, while the load was controlled by the height from the sample with/without the coverage of the SiOC/CPI cover window. We evaluated the performance of the TFHs before and after the impact pushing test, as shown in Figure 6c. The TFH without a cover window clearly showed a decreased saturation temperature and abnormal operation of the electrode region, as shown in the IR image (inset of Figure 6c). On the other hand, the TFHs covered by the SiOC/CPI cover window maintained their performance without degradation. Regardless of the thickness of the SiOC HC layer, the TFHs with a SiOC/CPI cover window functioned well, unlike the non-impacted devices.

Furthermore, the impact protection properties of SiOC/CPI films were examined with TFT devices. Figure 7a shows the fabrication procedure of the InZnSnO (IZTO)-based TFT devices. The TFT devices were fabricated on thermally oxidized 200 nm-thick SiO_2_-coated p^++^ Si substrates. The SiO_2_ and p^++^ Si were used as a gate insulator and global gate electrode, respectively. A 30 nm-thick indium zinc tin oxide (IZTO) channel layer was deposited by the RF sputtering of IZTO target, and a 100 nm-thick ITO S/D electrode layer was deposited using the DC sputtering of the ITO target. Both the channel and S/D patterns were defined via invar metal shadow masks. Figure 7b–d represent the drain current (*I*_d_)–gate voltage (*V*_g_) characteristics of the TFT devices at a constant drain voltage (*V*_d_) of 10 V. The *I*_d_–*V*_g_ curves before the impact pushing test show traditional n-type switching properties (turn on at positive *V*_g_, and turn off at negative *V*_g_). The gate leakage currents of the as-fabricated TFT devices are below 10 nA in the *V*_g_ of −10 to 50 V. When impact is applied to the TFTs without the cover window of the SiOC/CPI film, the *I*_d_–*V*_g_ curves move to the positive direction of *V*_g_, and the on-state current at a *V*_g_ of 50 V significantly deteriorates by about ten-fold. Additionally, when a high *V*_g_ is applied to the TFTs the gate leakage current drastically increases. In contrast, when TFTs are covered with SiOC/CPI films, as shown in Figure 7c,d, the *I*_d_–*V*_g_ curves are almost the same, even after the impact pushing test. Though the TFTs maintained their original *I*_d_–*V*_g_ characteristics, the gate leakage current was slightly increased. The results of the impact pushing tests of the TFHs and TFTs confirm that the R2R-sputtered SiOC films on the CPI substrate effectively protect the flexible devices from damage from external impacts.

## 4. Conclusions

In this paper, we report the development of flexible and transparent SiOC HC films deposited via a reactive R2R sputtering system on CPI substrates for use as flexible cover windows. The chemical and optical properties of the R2R-sputtered SiOC film were investigated to determine their potential as HC material. XPS analysis found that the SiOC HC layers were uniformly deposited on the CPI. Regardless of the thickness of the SiOC HC layer, the SiOC/CPI films showed a high optical transmittance of above 87% in the wavelength range of 400 to 800 nm. In addition, we performed mechanical flexibility tests, such as bending, folding, twisting, and rolling tests, to determine the flexibility of the R2R-sputtered SiOC films. Both the 100 and 300 nm-thick SiOC-coated CPI films showed outstanding mechanical flexibility at inner/outer bending, twisting, and rolling tests, without the generation of surface cracks. However, after the 100,000 cycles of the outer folding test, the 300 nm-thick SiOC-coated CPI films showed surface cracks due to high strain. Furthermore, the scratching resistive characteristics of the SiOC HC layers were examined by the standard pencil hardness test and steel wool scratching test. Finally, the impact protection ability of the SiOC films was investigated by applying an impact to the TFHs and TFTs. These tests revealed that our transparent and flexible SiOC/CPI displayed outstanding scratch resistance and impact protection characteristics as cover windows for foldable and flexible displays.

## Figures and Tables

**Figure 1 micromachines-12-00233-f001:**
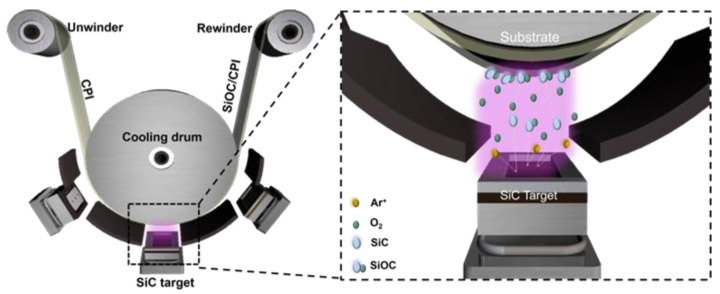
Schematics of the reactive sputtering process to deposit silicon oxycarbide (SiOC) films on colorless polyimide (CPI) substrates using a lab-scale roll-to-roll (R2R) sputtering system at room temperature (RT).

**Figure 2 micromachines-12-00233-f002:**
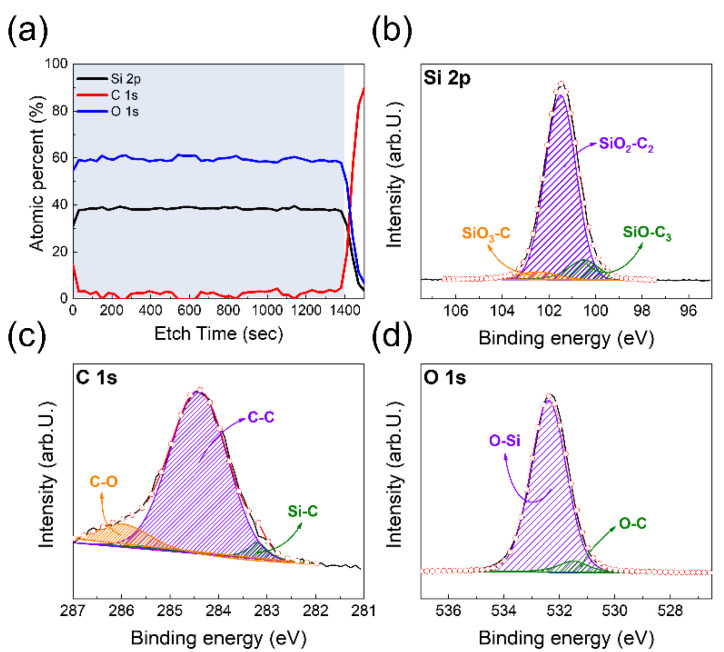
(**a**) XPS depth profile of R2R-sputtered SiOC hard coating (HC) film as a function of etch time. XPS spectra of (**b**) Si 2*p*, (**c**) C 1*s*, and (**d**) O 1*s* core level in reactive R2R-sputtered SiOC films.

**Figure 3 micromachines-12-00233-f003:**
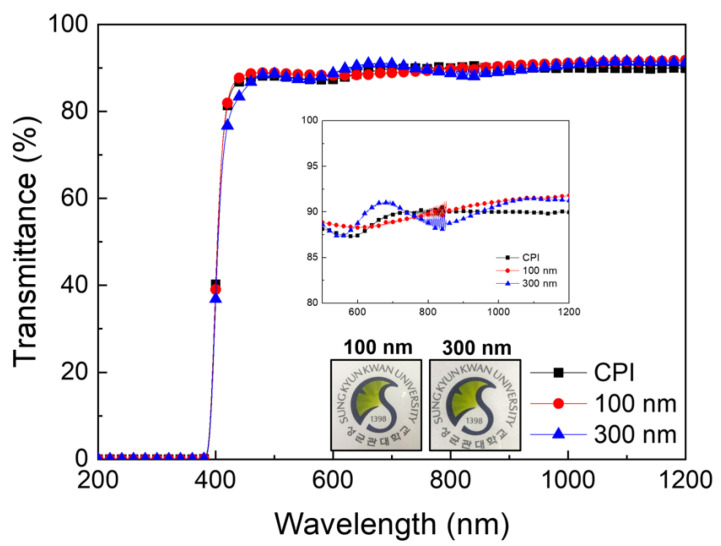
Optical transmittance of the bare CPI substrate and 100 and 300 nm-thick SiOC-coated CPI films. Insets show the enlarged spectra and pictures of the university logo through the transparent SiOC/CPI samples.

**Figure 4 micromachines-12-00233-f004:**
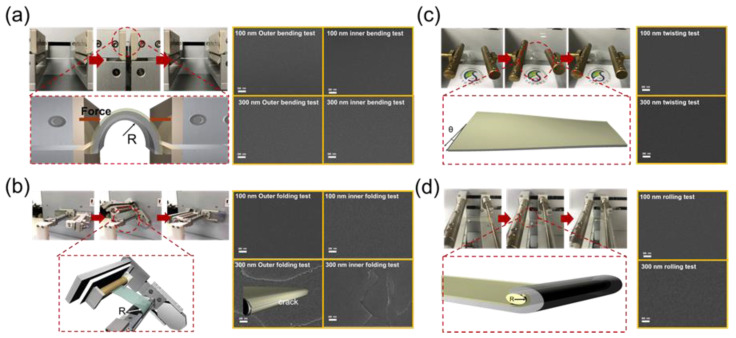
Schematic procedures of the flexibility test and surface field emission scanning electron microscopy (FE-SEM) images of the SiOC/CPI samples after the repeated (**a**) bending (static folding), (**b**) folding (dynamic folding), (**c**) twisting, and (**d**) rolling tests.

**Figure 5 micromachines-12-00233-f005:**
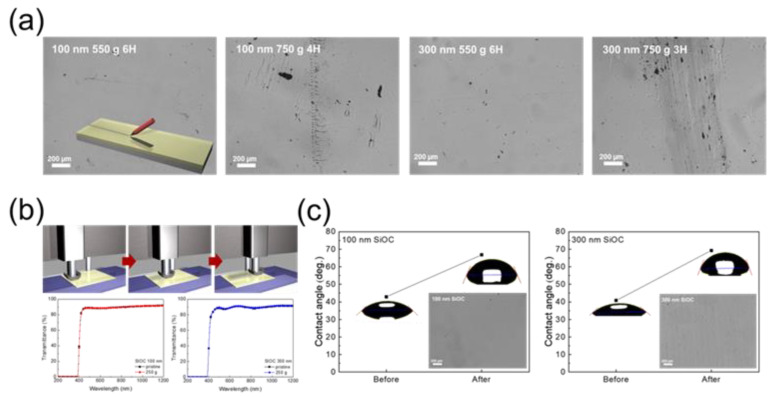
(**a**) Optical microscopy (OM) images of 100 and 300 nm-thick SiOC HC samples after the pencil hardness test under loads of 550 and 750 g. (**b**) Schematics of the steel wool scratching test and the transmittance of the SiOC/CPI samples before and after the steel wool test. (**c**) Change in contact angle of the SiOC/CPI samples before and after the steel wool test.

**Figure 6 micromachines-12-00233-f006:**
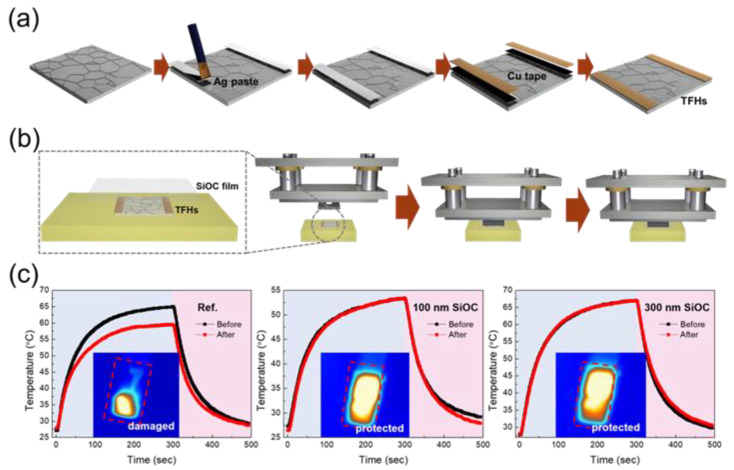
(**a**) Fabrication procedure of the thin-film heaters (TFHs) using Ag network electrode. (**b**) Steps of the impact pushing test. (**c**) Comparison of the TFH performance to investigate the impact protection ability of the SiOC/CPI films. Temperature profile of the TFHs was obtained by applying direct current (DC) power. Inset is the infrared (IR) image of the TFHs.

**Figure 7 micromachines-12-00233-f007:**
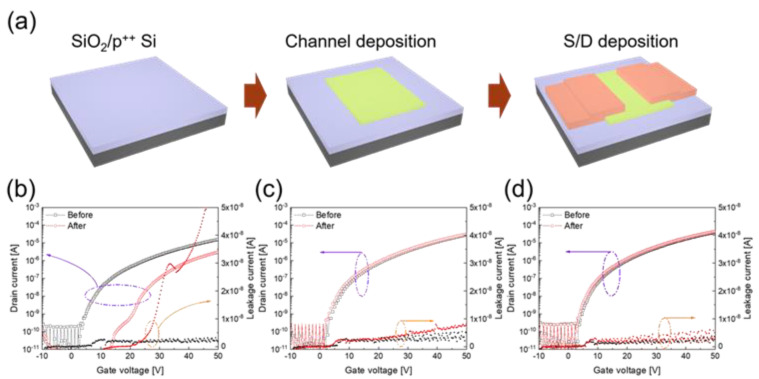
(**a**) Fabrication procedure of the TFTs using sputtered metal-oxide channel and electrode. I_d_–V_g_ characteristics of the TFTs before and after the impact pushing test (**b**) without SiOC/CPI film, and with a cover of (**c**) 100 nm-thick SiOC/CPI and (**d**) 300-nm thick SiOC/CPI films.

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
