# Peer review of "Transparent and Flexible SiOC Films on Colorless Polyimide Substrate for Flexible Cover Window"

_micromachines, 2021, doi:10.3390/mi12030233_

Round 1
Reviewer 1 Report
In this article the authors fabricated transparent and flexible silicon oxycarbide (SiOC) hard coating (HC) films on colorless polyimide substrate to use as cover window films for flexible and foldable displays by using reactive roll-to-roll (R2R) sputtering system at room temperature.
The article is interesting . However, I would suggest the authors to make amendements to this article.
- The authors should highlight in the introduction the novelty and the importance of their work.
- The authors should express the intensity as arb. U.
- It could be useful to add additional information about the index of refraction of the films.
Author Response
Reviewer #1:
1.The authors should highlight in the introduction the novelty and the importance of their work.
As reviewer mentioned, we added the novelty and importance of out work in the introduction part. We deposited a SiOC hard coating (HC) materials on CPI substrate for cover window of foldable display applications. Even some researchers have reported about the SiOC as a hard coating materials, there is no literature about SiOC hard coating films deposited by roll-to-roll (R2R) sputtering process. The R2R sputtering process can make possible to fabricate large-scale and uniform films. In this work, we deposited SiOC films by sputtering of SiC target with oxygen ambient. Our SiOC/CPI films showed superior optical transmittance about 87% and good mechanical felxibility under various deformation mode, such as bending, rolling, folding, and twisting. Furthermore, the SiOC/CPI fims have high resistance to scratching test and can protect the devices which are located under the films in impact pushing tests. These results shows that our SiOC/CPI films are suitable HC films for cover window of foldable display applications. We believe that our R2R sputtered SiOC/CPI films enable to apply for foldable display applications as a cover window.
2.The authors should express the intensity as arb. U.
As reviewer suggested, we modfied the unit “a.u.” to “arb. U.” in Fig. 2b-d.
3. It could be useful to add additional imformation about the index of refraction of the films
As reviewer suggested, we added additional imformation about the index of refraction of our SiOC/CPI films in Result and Disscusion part. According to the previous reports about the index of refraction of SiOC, the SiOC has an index of refraction about 1.5~1.8. Memon et al. reported that the index of refraction of SiOC are affected by the ratio between Si-C bond area and Si-O bond area. Also, Yoon el al. raised the transmittance of cover window films by adjusting the index of refraction. We anticipate that the optical properties of our SiOC/CPI cover window films can be also improved based on the previous reports.
Reviewer 2 Report
Han-ki KIM et al. has fabricated transparent engineering for application in cover window. I think the topic is very interesting and will have a potential application in transparent PV or other NIR materials applications. I recommend it for publication before one small thing addressed.
1. For the impact pushing test, it should be displayed more meticulously for the reader convenience.
Author Response
Reviewer #2:
- For the impact pushing test, it should be displayed more meticulously for the reader convenience.
As reviewer sugested, we added more detail information about impact pushing test in Materials and Method part. We added the fabrication methods of TFHs and TFTs and how we measured the change of performace of devices.
<Fabrication of TFHs>
Ag network-based TFHs were fabricated using a conductive Ag network films. Then, Ag paste was coated on the side of Ag network film with a dimension of 25 mm × 25 mm, to make a contact terminal for applying DC voltage. Finally, the contact region was covered with conductive Cu tape. The schematics of fabrication process are shown in Figure 6a.
<Fabrication of TFTs>
The TFT devices were fabricated on thermally oxidized 200 nm thick SiO2 coated p++ Si substrates. The SiO2 and p++ Si were used as a gate insulator and global gate electrode, respectively. A 30 nm thick indium¬ zinc tin oxide (IZTO) channel layer was deposited by RF sputtering of IZTO target, and a 100 nm thick ITO S/D electrode layer was deposited by using DC sputtering of ITO target. Both channel and S/D patterns were defined via invar metal shadow masks. The schematics of fabrication process are shown in Figure 7a.
<Impact pushing test>
5 N of impact were applied to prepared TFHs and TFTs wih or without the coverage of SiOC/CPI films using impact pushing tester (ET‐126‐4, Labworks Inc.). After the 10 cycles of impact pushing tests, the device performance of TFHs and TFTs were compared.
<Evaluation of TFHs>
Through the contact region of TFHs, a constant DC voltage of 3 V was applied to TFHs. Then, the temperature and IR thermal image of the TFHs were monitored using a thermocouple and IR camera. First, 3 V of DC voltage was applied for 300 sec. Then, the applying DC voltage was removed after 300 sec. When the DC voltage was applied to TFHs, the temperature of TFHs was increased and cooled down after 300 sec. We compared the highest temperature of TFHs before and after the impact pushing tests.
<Evaluation of TFTs>
The drain current-gate voltage (Id-Vg) characteristics of TFTs were evaluated using a semiconductor parameter analyzer (4200-SCS, Keithley). The Vg were swept -10 to 50 V. Finally, we compared the change of drain current and gate leakage current as a function of gate voltage.
Round 2
Reviewer 1 Report
I recommend the publication of this article.